# OpenCell: A low-cost, open-source, 3-in-1 device for DNA extraction

**Aryan Gupta**[1], **Justin Yu**[2], **Elio J. Challita**[3,4], **Janet Standeven**[3,5], **M. Saad Bhamla**[3]*

**1** School of Electrical & Computer Engineering, Georgia Institute of Technology, Atlanta, GA, United States of America, **2** School of Biological Sciences, Georgia Institute of Technology, Atlanta, GA, United States of America, **3** School of Chemical & Biomolecular Engineering, Georgia Institute of Technology, Atlanta, GA, United States of America, **4** George W. Woodruff School of Mechanical Engineering, Georgia Institute of Technology, Atlanta, GA, United States of America, **5** Lambert High School, Suwanee, Georgia, United States of America

* saadb@chbe.gatech.edu

## Abstract

High-cost DNA extraction procedures pose significant challenges for budget-constrained laboratories. To address this, we introduce OpenCell, an economical, open-source, 3-in-1 laboratory device that combines the functionalities of a bead homogenizer, a microcentrifuge, and a vortex mixer. OpenCell utilizes modular attachments that magnetically connect to a central rotating brushless motor. This motor couples to an epicyclic gearing mechanism, enabling efficient bead homogenization, vortex mixing, and centrifugation within one compact unit. OpenCell's design incorporates multiple redundant safety features, ensuring both the device's and operator's safety. Additional features such as RPM measurement, programmable timers, battery operation, and optional speed control make OpenCell a reliable and reproducible laboratory instrument. In our study, OpenCell successfully isolated DNA from *Spinacia oleracea* (spinach), with an average yield of 2.3 μg and an A260/A280 ratio of 1.77, demonstrating its effectiveness for downstream applications such as Polymerase Chain Reaction (PCR) amplification. With its compact size (20 cm x 28 cm x 6.7 cm) and lightweight design (0.8 kg), comparable to the size and weight of a laptop, OpenCell is portable, making it an attractive component of a 'lab-in-a-backpack' for resource-constrained environments in low-and-middle-income countries and synthetic biology in remote field stations. Leveraging the accessibility of 3D printing and off-the-shelf components, OpenCell can be manufactured and assembled at a low unit cost of less than $50, providing an affordable alternative to expensive laboratory equipment costing over $4000. OpenCell aims to overcome the barriers to entry in synthetic biology research and contribute to the growing collection of frugal and open hardware.

**Data Availability Statement:** All data can be found on GitRepo (https://github.com/bhamla-lab/OpenCell).

## Need for accessible and affordable DNA preparation hardware

In the rapidly evolving field of synthetic and molecular biology, the ability to efficiently isolate and purify nucleic acids is a fundamental requirement. However, the high cost of DNA

**Funding:** M.S.B. acknowledges funding support from NIH Grant R35GM142588; NIGMS SEPA Grant R25GM142044; NSF Grants MCB-1817334; and the Open Philanthropy Project. The funders had no role in study design, data collection and analysis, decision to publish, or preparation of the manuscript.

**Competing interests:** The authors have declared that no competing interests exist.

extraction procedures and the necessary equipment often poses significant challenges, especially for budget-constrained laboratories and educational institutions [1]. This financial barrier can limit the progress of research and hands-on STEM education, particularly in low-and-middle-income countries (LMICs).

Current extraction methods, while varied, share a common principle: they first rupture the cell membrane and suspend intracellular material in a homogeneous mixture, then capture the desired product in a silica-based filter, and finally elute the isolate from any impurities [1]. These steps necessitate several instruments, including a bead homogenizer, a vortexer, and a microcentrifuge. Bead homogenization, a common method of cell lysis at the laboratory scale, rapidly agitates high-density beads within a cell or tissue suspension. The applied shear forces break down cell membranes and expose intracellular material [1, 2]. However, even budget options for bead homogenizers can be prohibitively costly, with prices starting at $1000. Moreover, after cell lysis, additional equipment is needed to isolate the desired product from the lysate. The most common silica-based filtration methods necessitate a microcentrifuge (costing upwards of $3000) and a vortex mixer (ranging from $250 to $500) to achieve complete sample purification (Tables 1 and 2). While universities and industry providers offer nucleic acid and protein extraction services at $15-$100 per sample, this option may not be accessible in remote labs for example in the Amazon rainforest, LMIC institutions, maker spacers or even at rural high schools [3–7].

To combat these challenges, an emerging movement of open and frugal hardware is leveraging 3D-Printing and innovative engineering to replicate the functionality of commercial molecular and synthetic biology tools at a fraction of their original cost and size without sacrificing performance [8–12]. Several attempts have been made to repurpose existing equipment for effective bead homogenization and DNA sample preparation. For example, Michaels and Amasino utilized a paint shaker to homogenize samples in a 96-well plate [13]. However, paint shakers are uncommon tools for most labs, and the cost of the one used in their experiments, at $1,649, exceeds that of laboratory bead homogenizers. In another study, a reciprocating saw was modified for plant DNA extraction, showing positive results and costing between $80 and $150 [14]. Similarly, Peck et al. describe the use of a repurposed battery-operated oscillating multi-tool (power tool from a hardware store) named the PortaLyzer, which could be assembled for under $200 and approximated the performance of vortexer-based lysis methods [12]. However, both the reciprocating saw and the PortaLyzer present safety hazards as they repurpose high-speed power tools, which lack protective mechanisms in case of mechanical failure and lack convenient safety off-switch. This calls for more accessible and safe hardware options suitable for both laboratory and educational settings. Further, these devices are singular in purpose and are insufficient for completing any DNA isolation procedure without also a centrifuge.

**Table 1. Comparative analysis of commercial bead homogenizers and OpenCell.** This table provides a comparative analysis of two commercial bead homogenizers: the BeadBug 3-Position Bead Homogenizer and the Bead Ruptor 4, with the OpenCell bead homogenizer [27]. The comparison focuses on key specifications such as speed, capacity, weight, timer range, dimensions, and price. The OpenCell platform stands out for its significantly lower cost, lighter weight, and longer timer range, despite having a slower operational speed. Further, OpenCell is the only device that can be completely battery-operated.

| Specification | BeadBug | Bead Ruptor 4 | OpenCell |
|---|---|---|---|
| Speed | 2800–4000 rpm | 1 m/s–5 m/s | 450–1000 rpm |
| Capacity | 3 tubes | 4 tubes | 4 tubes |
| Weight | 2.2 kg | 16.5 kg | 0.8 kg |
| Timer Range | 3 sec–3 min | 1 sec–5 min | 15 sec to 10 min (adjustable) |
| Dimensions | 17 cm x 21 cm x 13.5 cm | 25.4 cm x 21.5 cm x 29.2 cm | 20cm x 28cm x 6.7cm |
| Price | $819 | $1995 | **$49.25** |

**Table 2. Comparative analysis of commercial centrifuges and OpenCell.** This table provides a comparative analysis of two commercial centrifuges: the Thermo Fisher Scientific mySPIN 6 Mini Centrifuge and the Eppendorf Centrifuge 5420, with the OpenCell centrifuge [28]. The comparison focuses on key specifications such as speed, capacity, weight, timer range, dimensions, and price. The OpenCell platform stands out for its significantly lower cost and inclusion premium features like adjustable speed and built-in timers, while maintaining a mid-range weight, footprint, and operating speed.

| Specification | mySPIN 6 | Centrifuge 5425 | OpenCell |
|---|---|---|---|
| Relative Centrifugal Force | 2000 × g | up to 21300 × g | up to 3000 × g |
| Capacity | 6 tubes | 24 tubes | 6 tubes |
| Weight | 0.74 kg | 16.5 kg | 0.8kg |
| Timer Range | N/A | 3 sec to 9:59 hrs | 15 sec to 10 min (Adjustable) |
| Dimensions | 10.4cm x 12.8cm x 15.3cm | 24cm x 39cm x 24cm | 20cm x 28cm x 6.7cm |
| Price | $672 | $3319 | **$49.25** |

Approaches to low-cost centrifugation, unlike bead homogenization, have been more recently explored. Devices inspired by whirligig toys, salad spinners, and egg-beaters that can achieve up to 30,000g of centrifugal force and up to 2ml sample tubes have been demonstrated [9, 15–17]. A key limitation of all these devices is their human-powered design, which although critical from a field diagnostics perspective, can require significant time and effort to operate. Another example is the Polyfuge, an Arduino-based benchtop microcentrifuge with speed and timer settings [10], which although more suitable for lab use, lacks any sensor for measuring motor speed and therefore any way to accurately control and modulate the applied centrifugal force, for DNA extraction protocols from different biological samples. Finally, the need for multiple bench top devices can be a limiting factor for field work and use in space or electricity constrained environments. Even in a lab setting, a single device can help to optimize a work-space and reduce the time required to move samples between devices.

## Design, fabrication and operation principles of OpenCell: A modular, open-source platform

The OpenCell platform, a cost-effective, open-source, and modular device, is designed to perform a variety of lab operations, including cell lysis, sample mixing, and centrifugation (Fig 1A). The total cost of the OpenCell platform, excluding batteries, is approximately under $50.00. This cost includes the components such as the Arduino Nano, brushless motor, electronic speed controller (ESC), display, buttons, potentiometer, DC power supply, Hall Effect sensor, bearings, wires, breadboard, fasteners, master switch, and the PLA filament for 3D printing (Table 3, S1 Table). For portable applications, a 3S 11.1V LiPo drone battery can be used for approximately $11.50.

The OpenCell platform is constructed using 3D-printed components, primarily made of Polylactic Acid (PLA) filament. This material was chosen for its user-friendly nature and compatibility with all filament-based Fused Deposition Modeling (FDM) 3D printers, such as the Creality Ender series or Prusia MK3. More robust materials like Polyethylene Terephthalate Glycol (PETG) or Acrylonitrile Butadiene Styrene (ABS) can be employed to enhance the device's resilience to high temperatures and wear [18–20]. The components were designed using Autodesk's Fusion360 and prepared for printing with Ultimaker Cura. All parts are limited to a maximum size of 200mm x 200mm x 50mm to ensure compatibility with most consumer 3D printers. Overhanging surfaces have been optimized to minimize support usage wherever possible. The components are assembled using m3 fasteners, minimizing the use of glue or solvents, which facilitates easy assembly, removal, or repair, making maintenance and potential repairs straightforward and user-friendly.

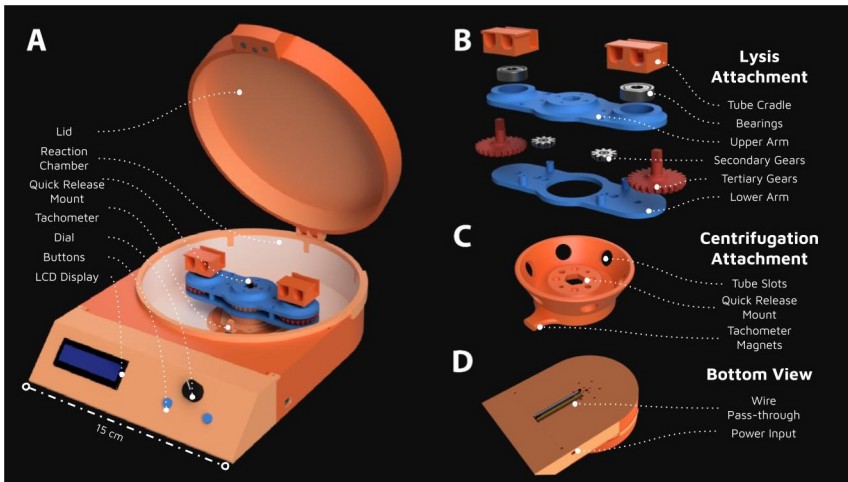

**Fig 1. Design and assembly of the OpenCell device.** Computer Aided Design (CAD) renderings of the fully assembled OpenCell device, showcasing its modular and user-friendly design. The device is constructed entirely from 3D-printed and off-the-shelf components at a unit cost of less than $50 (Table 3). (A) The fully assembled OpenCell device with the lysis attachment in place. An LCD display, potentiometer, and two momentary switches are used to control control the motor speed and run time. The device weighs approx 0.8 kg and is 20cm x 28cm x 7cm(B) An exploded view of the lysis attachment, highlighting the epicyclic gear system and the 3D-printed components. The unique orbiting motion generated by this attachment induces a reciprocating motion within the tubes necessary for cell lysis. (C) The centrifugation attachment, demonstrating the device's versatility and adaptability for different laboratory procedures. (D) A bottom-facing view of the device, providing a comprehensive perspective of the OpenCell's compact and efficient design.

The OpenCell platform is powered by an Arduino Nano microcontroller, which controls the electronic inputs and outputs of the device. The powertrain of the device is driven by a 2206 2600kv brushless motor, regulated by a 30A Electronic Speed Controller (ESC). This setup ensures ample power for the device's operations (S1 File). The OpenCell platform can be powered by any 6–12 V DC power supply; we recommend a supply capable of delivering at least 2 amps of sustained current for similar performance. However, the OpenCell platform is versatile and can be powered with a 3s 11.1 V LiPo battery for cordless operation. In our testing protocol, which consists of 5-minute operation intervals followed by 1-minute rest periods to prevent overheating, a 2200mAh Lithium Polymer (LiPo) battery could power the lysis attachment for 60 minutes and the centrifuge attachment for over 90 minutes. This duration

**Table 3. Market cost of purchased components of opencell.** This table provides a high-level breakdown of the component costs of the OpenCell platform. The prices reflect the market costs in 2023. The total cost is approximately under $50. A more detailed bill of materials can be found in the supplemental materials (S1 Table).

| Component | Description | Price |
|---|---|---|
| Nano v3 | Arduino Compatible microcontroller | $4.25 |
| Brushless Motor | 2205 or 2206 Drone Motor | $8.00 |
| ESC | Electronic Speed Controller | $9.50 |
| Misc. Electronics | Sensors, Display, Wires, Etc. | $9.00 |
| Misc. Hardware | Fasteners, Bearings, Etc. | $4.25 |
| Power Supply | 3s LiPo or 12v 3amp DC | $9.50 |
| Filament Cost | Approximately 425g of PLA | $4.25 |
| **Total** | | **$49.25** |

allows for the processing of 8 sample tubes or approximately 16 ml on a single charge, providing flexibility for both stationary and portable use.

Wires from the main chamber are securely routed to the electronics chamber via a wire-pass-through channel, reducing the risk of disconnection or damage (Fig 1D). All electrical connections can be made via a solderless breadboard that fits within the electronics chamber of the device. This design allows even minimally trained users to assemble and operate the device (S2 and S3 Videos). For enhanced stability, especially during transportation (lab-in-a-backpack), soldering connections onto a protoboard is recommended.

## Cell lysis using epicyclic gearing

The OpenCell platform's cell lysis attachment, also used for sample mixing, is a key innovation that employs an epicyclic gear system (Fig 1B). This system is designed to convert the rotational torque output from the brushless DC motor into the rapid reciprocating motion necessary for cell lysis.

The epicyclic gear system consists of a stationary central gear and two tertiary gears of identical diameter, along with two secondary idler gears of arbitrary diameter (Fig 1B). These gears are constrained by an upper and lower arm, with rotary bearings at each rotational joint. This design reduces friction and minimizes wear on the 3D-printed components, making it highly suitable for 3D printing. Furthermore, due to the inertial symmetry of the epicyclic gear train, a consistent amount of torque is required to operate throughout an entire rotation, effectively reducing the stress exerted on the powertrain and 3D-printed structure.

The operation of the cell lysis attachment is best understood by examining the spatio-temporal dynamics of beads in the tube during cell lysis operation. As the DC motor applies torque to the central arms, the entire attachment rotates about the fixed main gear with an angular velocity, denoted as $\omega$ (Fig 2). This rotation causes the torque from the main gear to be inverted via the secondary gears before reaching the tertiary gears. As a result, the tertiary gears rotate about their central axis with an angular velocity of $-\omega$. This means that the tertiary gears experience no relative angular velocity, and their orientation is mechanically stabilized to that of the central gear.

When 2ml tubes are affixed to the tertiary gear, the internal contents, which include the beads, experience a centrifugal force proportional to their rotation about the central gear. This is the same principle that a centrifuge operates on. However, because the tubes also rotate about the center of the tertiary gear, the direction of the centrifugal force relative to the orientation of the tube will also rotate with angular velocity $\omega$. Due to the elongated nature of the tube, bead movement is constrained to a single dimension. Thus, the bead acceleration within the tube is best described as a sinusoidal path bound by the orientation of the arm and with a magnitude proportional to $\omega$.

The spatiotemporal dynamics of this system are observed using a high-speed camera recording at 4025 fps (S1 Video). We identify two consecutive regimes within each rotational cycle of the attachment (Fig 2A and 2B). The first translation stroke begins at the peak of the rotation, where beads accelerate upwards towards the cap of the tube. As the attachment continues to rotate, this stroke continues until the beads collide with the cap of the tube, initiating the first impact stroke. During the impact stroke, the beads experience negligible acceleration until the attachment approaches the base of the rotation, where the second translation stroke begins. Here, the beads begin to accelerate downwards towards the bottom of the tube. Finally, the beads collide with the bottom of the tube and initiate the second impact stroke, where they experience negligible acceleration until the cycle repeats.

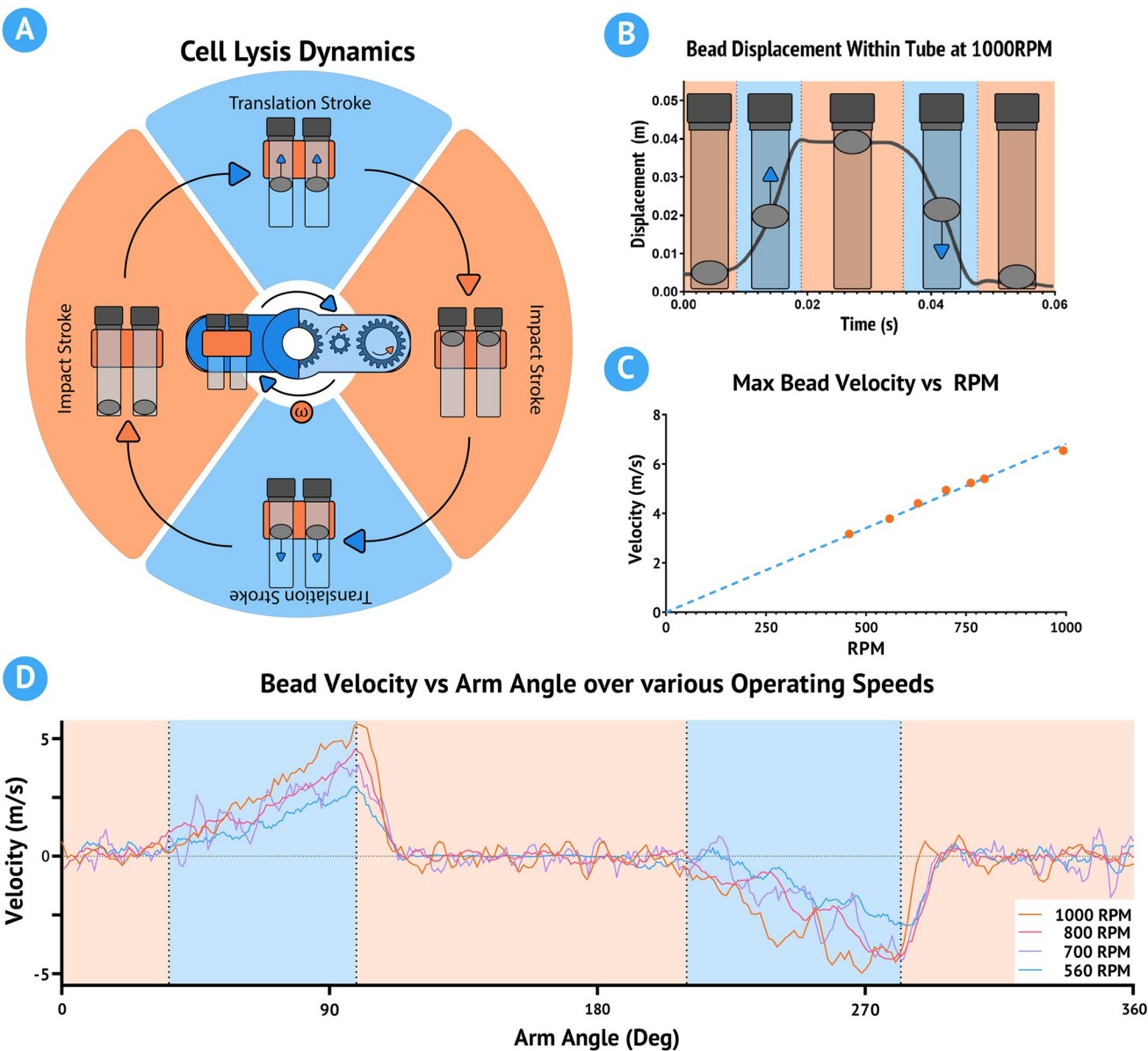

**Fig 2. Spatiotemporal dynamics and performance of the cell lysis module.** (A) Diagrammatic representation of the spatiotemporal dynamics of the cell lysis module, illustrating the motion of beads within the tube during operation. (B) Graphical representation of bead displacement overlaid with the measured bead displacement at an operating speed of 1000 RPM. (C) Graph illustrating the linear relationship between the bead velocity and the module's RPM, demonstrating the module's performance consistency across different speeds. (D) Graph showing the measured bead velocity with respect to the orientation of the arm across multiple operating speeds, highlighting the consistency of bead motion. The orange zones in parts A, B, and D represent impact strokes, while the blue zones represent transition strokes. Bead velocity is computed as the derivative of displacement, smoothed with an 8-point moving average filter. See (S1 Video) for a high speed video of the lysis attachment in motion.

Using the open-source software 'Tracker', we measure the bead dynamics at a variety of speeds and show that the peak bead velocity before impact has a linear relationship with the operating speed of the device (Fig 2C and 2D) [21]. The OpenCell bead-mill attachment has an operating range of 300–1200 RPM, or 10–40 collisions per second, with a maximum bead velocity of 6.5 m/s. The tube cradles are designed to hold 2 tubes per side and use a locking cover to maintain compatibility with most 2ml tube designs.

## Sample mixing and centrifugation modalities

The OpenCell platform, beyond its primary function of cell lysis, is versatile enough to perform sample mixing and centrifugation. The cell lysis attachment, typically used for bead homogenization, can be repurposed for sample mixing by simply excluding the beads. The same forces that drive bead homogenization also facilitate the effective mixing of samples. For sensitive mixtures, the device's operating speed can be adjusted to reduce the peak liquid velocity during each rotation, ensuring gentle and thorough mixing (Fig 2C).

In addition to mixing, the platform features a custom-designed centrifugation attachment that can be quickly swapped with the bead homogenization module using magnets to snap securely in place (Fig 1C). This attachment, designed to accommodate six sample tubes at a 60-degree angle, can safely accelerate samples up to $3000 \times g$ (Fig 1). While this is only 15% of what conventional benchtop centrifuges can achieve (Table 2), our testing demonstrates that only $2500 \times g$ is sufficient for DNA extraction with some protocol adjustments (discussed in detail below), further underscoring the OpenCell platform's adaptability in various laboratory procedures.

The OpenCell platform, with its cell lysis, sample mixing, and centrifugation attachments, exemplifies a modular design. A magnetic interface enables swift and secure swapping of these attachments, eliminating the need for cumbersome fasteners and reducing the risk of component loss (Fig 1). This design ensures stability and safety during high-speed operations. The platform's modularity is further highlighted by the inclusion of a template CAD model, facilitating the development of new attachments (S2 File). This design approach underscores the OpenCell platform's adaptability, making it a versatile tool for diverse laboratory procedures.

## Safety measures and user-friendly features

We recommend wearing lab coats, safety goggles, and additional personal protective equipment when using OpenCell. Like any spinning motor, this device produces significant noise, necessitating some form of hearing protection.

OpenCell can accelerate samples to over $3000 \times g$, over 6000 RPM, given an adequate power supply. To minimize the risk of damage to the device and ensure user safety, we have implemented both preventative and mitigating safety features in the OpenCell platform.

First, a software-based safety check, facilitated by A3144 Hall Effect sensors, inhibits motor operation if the enclosure lid is improperly seated or removed during operation. Second, a software timer limits any continuous operation to 10 minutes to reduce risk of components overheating. Third, a master power switch fully disables the device if software controls become unresponsive. Last, the device features a reinforced chamber and lid to protect the user in the event of a catastrophic failure (Fig 1A).

By modeling the Centrifugation attachment as a simple disk with a 70mm radius, 10mm thickness, and loaded weight of 40g, we estimate its total kinetic energy at approximately 14J. The impact resistance of 3D printed PLA, measured between 10 and 18 $kJ/m\hat{2}$ depending on print parameters [22, 23], suggests that in the event of a part failure, PLA would have sufficient impact resistance to safely absorb the energy across its large area. To demonstrate OpenCell's safety, we conducted various tests, including operating the the Centrifugation and Cell Lysis attachment in imbalanced conditions (see S4 Video). In both scenarios, the device showed no observable damage. To simulate a worst-case scenario, we deliberately damaged a centrifugation attachment using a saw until it could be crushed by hand. Running this weakened attachment until its structure failed and explorded, we found that all fragments remained contained within the chamber, causing no visible damage to its structure. During these safety tests, we noted OpenCell's tendency to shift due to vibrations. We recommend applying adhesive or

rubber strips to the base of the OpenCell to provide a high-friction surface and reduce any shifting.

In addition to these safety measures, the OpenCell platform is designed for ease of use, accessibility, and adaptability. It features a simple interface with an I2C-enabled 16x2 LCD, a potentiometer, and momentary switches (see Fig 1). Similar to commercial devices, users can define the speed and duration of operation using this interface. An additional Hall Effect sensor within the attachment chamber provides precise speed measurements by tracking pulses from magnets embedded within the attachments. Users can view the measured attachment speed, desired speed, current motor power, and timer on the LCD display. Additionally, users can adjust the motor power in real-time to fine-tune the device speed. A version of the code enables a closed-loop PID controller to intelligently maintain attachment speed to within ±1% of the desired setting (S1 and S2 Files), making the OpenCell platform a reliable and user-friendly tool for various lab operations.

## Optimization and characterization of OpenCell for DNA extraction

We next optimize and characterize the OpenCell platform for DNA extraction from *Spinacia oleracea* (spinach) using the commercially avaliable Quiagen DNeasy plant pro kit shown to be effective in the literature [21, 24]. This optimization aligns OpenCell's performance with commercial protocols that assume the use of commercially available and calibrated equipment (S3 File). The DNA extraction process divides into four steps: lysis, binding, washing, and elution, each utilizing different OpenCell attachments as shown in (Fig 3A). We evaluate the performance of OpenCell by quantifying DNA yield and purity using a NanoDrop or spectrophotometer, with pure DNA having an A260/A280 ratio of 1.8 [25, 26]. For detailed protocols and considerations for these experiments, see S3 File.

We optimize the operational parameters for the cell lysis (bead-homogenization) and centrifugation modules of OpenCell. Initial testing establishes an operating speed range for the cell lysis module from 425 to 1000 RPM, with an optimal range of 725–850 RPM determined based on DNA yield and purity from three independent trials (Fig 3C). Similarly, an optimal operating duration range of 120–180 seconds was determined for the cell lysis attachment from three independent trials(Fig 3B). For the centrifugation module, an optimal factor of 1.5x the protocol recommended duration is determined for efficient DNA isolation from three independent trials, at $2500 \times g$ (Fig 3D).

**Comparative performance of OpenCell as a 3-in-1 DNA extraction tool.** We next contextualize the performance of the OpenCell system by comparing it with commercial devices. For this purpose, we define three experimental groups: OpenCell, commercial, and control. In the OpenCell group, we process samples exclusively using the optimized parameters derived from our prior characterization experiments. This includes the utilization of the OpenCell system for all cell lysis, centrifugation, and sample mixing steps.

The commercial group adheres to the predefined DNA extraction protocol (S3 File) provided with a commercial reagent kit, using the following commercial devices: Eppendorf 5245 24-tube microcentrifuge, BeadBug 3 Position bead homogenizer, and benchtop vortex mixer (Tables 1 and 2). This group serves as a positive control and represents an ideal scenario for a DNA extraction. The control group undergoes the entire process without the use of any mechanical devices, representing a negative control where no usable result is expected.

The results, shown in (Fig 4A), demonstrate that the OpenCell system consistently generates DNA extracts with both satisfactory yield and A260/A280 values well within the usable range. Further, when compared to the commercial group, the average OpenCell DNA yield is within 5% of the commercial DNA yield, and the average A260/A280 ratio is within 3%. The

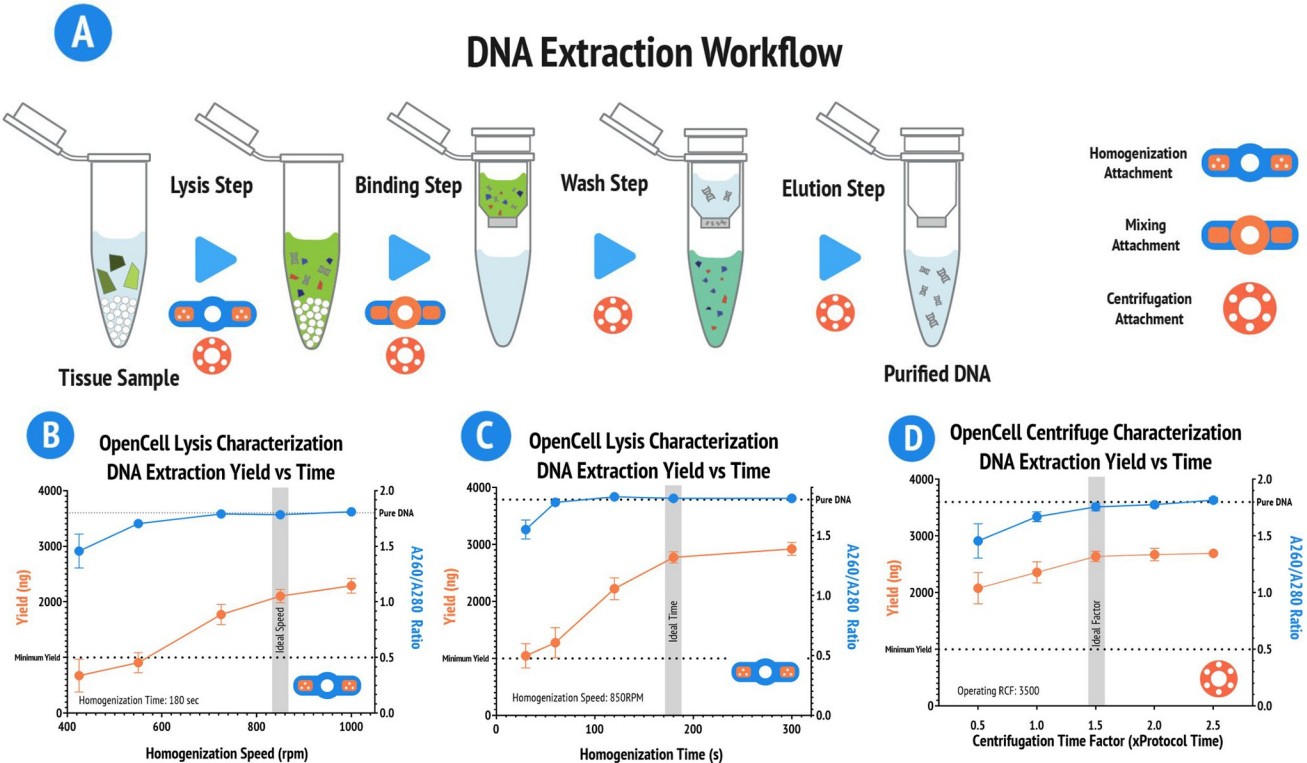

**Fig 3. Characterization and optimization of OpenCell for DNA extraction. (A)** Diagram demonstrating the basic steps of a DNA extraction workflow, with corresponding OpenCell attachments. The process includes lysis, binding, washing, and elution. **(B)** Results of speed modulation experiments for the cell lysis (bead-mill) module with a fixed operating time of 180 seconds (n = 3). The optimal operating speed range was determined to be 725–850 RPM. **(C)** Results of time modulation experiments for the cell lysis module with a fixed rpm of 850RPM (n = 3). The optimal operating duration range was determined to be 120–180 seconds. **(D)** Results of time modulation experiments for the centrifugation module with a fixed operating RCF of 2500 × g (n = 3). An optimal factor of 1.5x the protocol recommended duration was determined for efficient DNA isolation.

most significant difference between the two groups is consistency: across eight independent trials the standard deviation for amount of DNA extracted for the OpenCell data set is 399.6 ng (17%), compared to 213.2 ng (9%) for the commercial group.

We then performed gel electrophoresis on the extracted DNA to verify the presence of *Spinacia oleracea* genomic DNA. The DNA sequence length was consistent and accurate between the OpenCell-extracted DNA and the commercially extracted DNA, indicating the success of the DNA extraction.

We further validate the utility of the OpenCell-extracted DNA through downstream analysis. We target the Ribulose bisphosphate carboxylase large chain (Rbcl) gene, a common marker in *Spinacia oleracea*, for this analysis. The extracted DNA undergoes PCR and the resulting product is analyzed using gel electrophoresis (S3 File).

The gel electrophoresis results reveal defined bands within the 500–600 bp range for each sample tested, indicative of successful PCR amplification (Fig 4B). This demonstrates that the DNA extracted using OpenCell is intact and of sufficient quality for downstream applications. These results underscore the potential of OpenCell as a viable tool for DNA extraction in molecular and synthetic biology research.

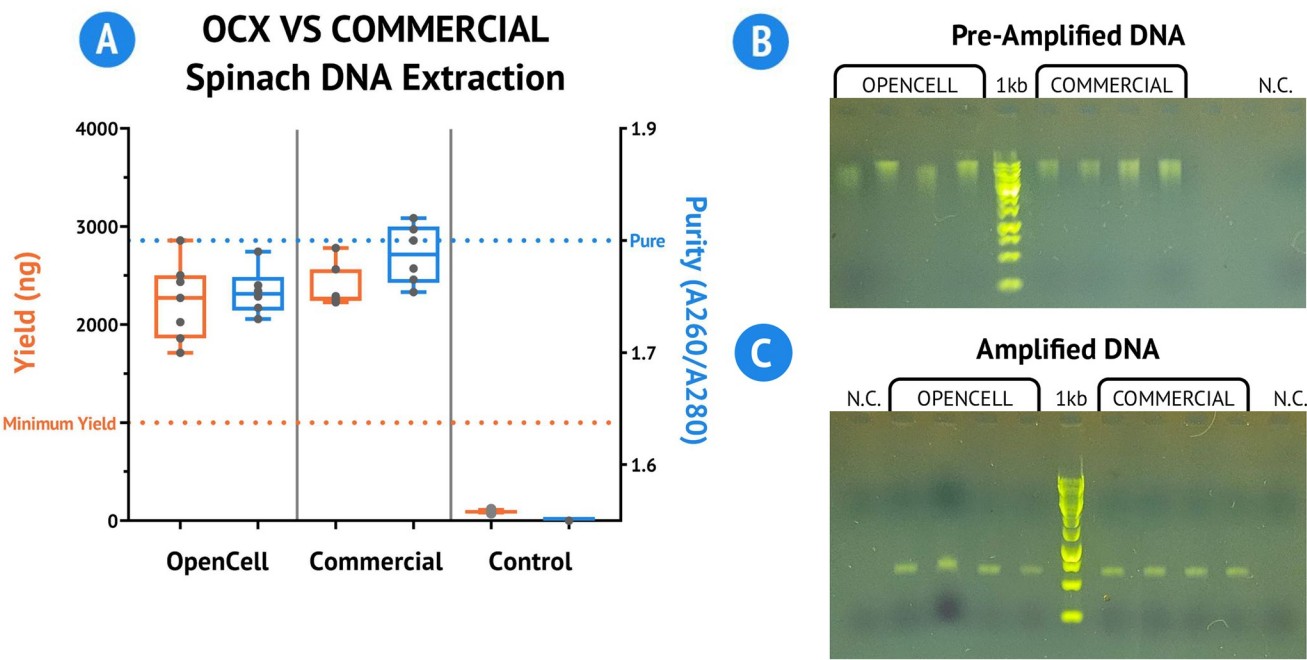

**Fig 4. Comparative performance of OpenCell and downstream validation of extracted DNA. (A)** Results of the DNA extraction experiment quantified using a NanoDrop Lite (n = 8). The average DNA yield and A260/A280 ratio of the OpenCell group were within 5% and 3% of the commercial group, respectively. **(B)** Gel electrophoresis results of the extracted DNA before PCR amplification (n = 4). Defined bands were consistent between OpenCell-extracted DNA and commercially extracted DNA. **(C)** Gel electrophoresis results following PCR purification procedure targeting the Rbcl gene, approximately 600bp long (n = 4). Defined bands within the 500–600 bp range were observed for each sample tested, indicating successful PCR amplification. In both the pre-amplified DNA gel and the amplified DNA gel, the ladder indicated is a 1kbp Log Ladder sourced from New England BioLabs, and the lane labeled N.C. contains a negative control.

## Advantages and potential limitations of OpenCell

OpenCell, with its cost-effectiveness and accessibility, represents a significant advancement in the democratization of synthetic biology laboratory equipment. At a unit cost of just less than $50, OpenCell is less than 1% of the cost of the tsahree commercial devices it emulates (Tables 1 and 2). The widespread availability of FDM 3D printers in educational and public maker spaces further enhances its accessibility, extending its potential use to budget-conscious labs in LMIC, high school science labs and beyond [29].

The device's multifunctionality, consolidating bead-homogenization, centrifugation, and sample mixing within a compact footprint, facilitates quick and easy function switching. This consolidation not only saves valuable lab space but also reduces the equipment required for fieldwork. OpenCell's portability is further enhanced by its ability to operate battery-powered for over 60 minutes, thanks to the use of consumer drone LiPo batteries and ESCs for motor control. This feature, combined with its backpack-friendly footprint and low weight (0.8kg), makes OpenCell particularly suitable for fieldwork or use in locations with limited access to electricity (Table 1).

However, the design of OpenCell does present some limitations. Its 3D-printed construction, while cost-effective, is less durable than commercial devices, particularly in high-stress components such as the motor hub and gears. Regular monitoring and maintenance of these components are necessary to prevent mechanical failure. We recommend the use of a polymer-safe lubricant, such as white lithium grease, to enhance component longevity. While OpenCell is designed to accommodate any 22mm diameter brushless motors, single motor

ESC, and power supply, any substitutions will directly impact the device's performance and must be rigorously evaluated.

FDM 3D printing, while accessible, is not a precise manufacturing method. Variations in dimensional accuracy between printers, filament brands, colors, materials, and printing parameters can impact the strength and characteristics of a component. Although the design includes large tolerances to accommodate these variations, slight modifications in scale may be necessary for components to interface smoothly. Additionally, any process defects for 3D printed components can have a substantial effect on their strength. See supplemental section for ideal print parameters and resources for identifying common 3D printing defects (S1 Table).

PID controllers can be difficult to tune and an unsuitable controller could damage the device's electronics. Further, the PID constants should be tuned separately for different power sources as they directly impact motor performance. For inexperienced users, an open-loop version of the program is also provided where the motor power is set manually during operation.

## Concluding remarks

OpenCell's open-source approach aligns with our commitment to promoting accessible science. We provide comprehensive information about every component, whether 3D-printed or off-the-shelf, along with any corresponding 3D models in the supplementary section (S1 Table, S2 File). A reference template is also included to facilitate the design of new attachments.

By successfully preparing samples for a DNA workflow, OpenCell demonstrates its capability to match commercial devices at a fraction of the cost. Its software features, such as speed control, programmable timers, and safety overrides, integrate seamlessly into a modern laboratory setting. As an addition to the growing branch of frugal science, OpenCell serves as a gateway to molecular and synthetic biology research and the democratization of science.

## Supporting information

**S1 Video. High speed recording.** 4025FPS High speed recording of OpenCell Lysis Attachment operating at 1000RPM.
(MP4)

**S2 Video. Assembly instruction video.** A video walk-through of assembling OpenCell and the two attachments.
(MP4)

**S3 Video. Operating instruction video.** A video walk-through of operating OpenCell safely.
(MP4)

**S4 Video. Stress testing videos.** Recordings of stress testing OpenCell under imbalanced load and worst case scenarios.
(MP4)

**S1 Table. Slicer settings.** The appropriate slicer settings for each respective part.
(PDF)

**S1 File. Supplemental figures.** Diagram of all electronic components used in OpenCell, attachment speed over time using PID controller.
(PDF)

**S2 File. Component files.** All required STL files and Arduino code. More information can be found on GitRepo (https://github.com/bhamla-lab/OpenCell).
(ZIP)

**S3 File. Manual.** Assembly Instructions, Operation Instructions, and Protocols.
(PDF)

**S1 Raw images. Raw images.** Uncropped, labeled, images of gels shown in Fig 4.
(PDF)

## Acknowledgments

We thank members of the Bhamla lab for useful discussions; Lambert iGEM for their support and collaboration; Sahil Jain and Saikishore Mettupalli for their help with initial designs.

## Author Contributions

**Conceptualization:** Aryan Gupta, Justin Yu, Janet Standeven.

**Data curation:** Aryan Gupta, Justin Yu, Elio J. Challita.

**Formal analysis:** Aryan Gupta, Justin Yu.

**Funding acquisition:** Janet Standeven, M. Saad Bhamla.

**Investigation:** Aryan Gupta, Justin Yu, Elio J. Challita.

**Methodology:** Aryan Gupta, Justin Yu, Elio J. Challita, Janet Standeven.

**Project administration:** Janet Standeven, M. Saad Bhamla.

**Resources:** Janet Standeven, M. Saad Bhamla.

**Software:** Aryan Gupta, Justin Yu.

**Supervision:** Elio J. Challita, Janet Standeven, M. Saad Bhamla.

**Validation:** Elio J. Challita, Janet Standeven, M. Saad Bhamla.

**Visualization:** Aryan Gupta, Justin Yu.

**Writing – original draft:** Aryan Gupta, Justin Yu, Elio J. Challita.

**Writing – review & editing:** Aryan Gupta, Justin Yu, Elio J. Challita, Janet Standeven, M. Saad Bhamla.

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
