## [Decision Letter · Decision Letter 0]

30 Oct 2023

PONE-D-23-30724OpenCell: A Low-cost, Open-Source, 3-in-1 device for DNA ExtractionPLOS ONE

Dear Dr. Bhamla,

Thank you for submitting your manuscript to PLOS ONE. After careful consideration, we feel that it has merit but does not fully meet PLOS ONE’s publication criteria as it currently stands. Therefore, we invite you to submit a revised version of the manuscript that addresses the points raised during the review process.

The reviewers are very positive about the manuscript and I agree, yet, they also raised some important questions  e.g., regarding the durability of the 3D-printed parts with the forces applied to it during centrifugation that need to be addressed in a revised version of the manuscript before it can be published. 

We look forward to receiving your revised manuscript.

Kind regards,

Sven Winter

Academic Editor

PLOS ONE

Journal Requirements:

"M.S.B. acknowledges funding support from NIH Grant R35GM142588; NIGMS SEPA Grant R25GM142044; NSF Grants MCB-1817334; CAREER IOS-1941933; and the Open Philanthropy Project. We thank members of the Bhamla lab for useful discussions; Lambert iGEM for their support and collaboration; Sahil Jain and Saikishore Mettupalli for their help with initial designs."

"M.S.B. acknowledges funding support from NIH Grant R35GM142588; NIGMS SEPA 308 Grant R25GM142044; NSF Grants MCB-1817334; CAREER IOS-1941933; and the 309 Open Philanthropy Project. The funders had no role in study design, data collection and analysis, decision to publish, or preparation of the manuscript."

Reviewers' comments:

Reviewer's Responses to Questions

**Comments to the Author**

1. Is the manuscript technically sound, and do the data support the conclusions?

Reviewer #1: Yes

Reviewer #2: Yes

2. Has the statistical analysis been performed appropriately and rigorously? 

Reviewer #1: Yes

Reviewer #2: N/A

3. Have the authors made all data underlying the findings in their manuscript fully available?

Reviewer #1: Yes

Reviewer #2: Yes

4. Is the manuscript presented in an intelligible fashion and written in standard English?

Reviewer #1: Yes

Reviewer #2: Yes

5. Review Comments to the Author

Reviewer #1: In this work, Gupta et al. present the design and characterization of an open-hardware DNA extractor lab instrument. Open-source lab hardware is a small but growing field, and this paper makes a valuable and high-quality contribution to this field. The authors’ instrument replicates the function of three common lab tools: a bead homogenizer, a centrifuge, and a vortex mixer. The design’s engineering elegance (the epicyclic gearing especially) and modular nature (using magnets to quickly swap parts for different functions) are particularly noteworthy. Additionally, the authors include a thorough characterization of the instrument’s performance. I enthusiastically support the publication of this manuscript in PLOS ONE once the authors address a few concerns below.

1. My most significant concern with the manuscript involves the safety of the instrument and any risk it might pose for the operator. Perhaps some elevated risk is acceptable and unavoidable in a laboratory environment, but in settings like classrooms and makerspaces (where the authors rightly suggest their instrument could find valuable applications), risk must be kept to an absolute minimum. The authors already have some thoughtful risk-mitigating features in their design, for example the Hall effect sensors used to detect premature opening of the lid while the unit is in operation and various software-based safety checks. However, for any tool involving a mass rotating at a high speed, the primary risks come from mechanical failure of the rotating mass (potentially sending high-speed debris into the surroundings) and the emergence of violent vibrations (from an unbalanced rotating load). The authors state that the “reinforced [3D-printed] chamber and lid… protect user in case of a catastrophic failure,” but this important claim is presented without any evidence. And while I trust the authors’ judgement of their own 3D-printed parts, there’s so much variability in the quality and durability of parts from different 3D printers; I can imagine scenarios where e.g. a poor-quality or poorly maintained printer, or a poor quality filament, leads to parts that are much weaker than the ones produced by the authors.

A quick back-of-the-envelope calculation to illustrate the possible risks: the authors’ instrument operates at up to 3500 g’s of relative centrifugal force. Assuming that each 2 mL tube weighs 3 grams, when spun at 3500 g’s, each tube would exert a force equivalent to 10 kilograms or 22 pounds of force on the 3D-printed frame holding the tube. Since there are four tubes mounted on the rotating frame, I think the total tensile force inside the frame would be quadrupled to around 88 pounds of force. That’s a lot of force – much more than I expect most of my own 3D-printed parts to endure without breaking.

I’d ask the authors to address these risks more directly in the paper. Some specific suggestions:

- Include this sort of rough force or stress analysis in the paper, so that readers can decide whether their own 3D printer is suitable for printing parts that must endure those sorts of forces and stresses.

- Include recommendations for personal protective equipment that should be worn when operating the instrument. At a minimum, eye protection is essential and isn’t mentioned in the manuscript.

- Probably the most common mistake when operating a centrifuge is not balancing the samples in the rotor. I’d like the authors to intentionally load and operate their instrument unbalanced (for example, load three tubes in the centrifuge attachment but all on the same side of the rotor, and load two tubes in the lysis attachment but both on the same side of the arm) and carefully observe the results and include them in the manuscript. My worry is that the magnetic mounts might not be able to endure the vibrations caused by an unbalanced attachment. If so, the use of magnetic mounts may need to be reevaluated. Even if the attachments remain attached in these tests, the effect of an unbalanced attachment on the overall instrument should also be tested. The lightweight 3D-printed instrument body might be slung around or even knocked off a table. If this happens, altering the design to accommodate a clamp for holding the instrument onto a table could mitigate this danger.

Addressing safety concerns directly will help make sure that the authors’ (very creative and useful!) tool doesn’t harm anyone, even if the chance of that happening is low.

2. The supporting material is excellent. One minor request, you might try to use video editing software (or even ImageJ) to increase the brightness of the S1 (high-speed) video. Also I’d suggest moving Table 3 from the main article text to the supplementary information and incorporating Amazon URLs into the table and not just listing them in the caption.

3. In the Conclusion, this sentence:

“By successfully performing a DNA workflow, from DNA extraction to polymerase chain reaction and gel electrophoresis, OpenCell demonstrates its capability to match commercial devices at a fraction of the cost.”

makes it sound like the OpenCell instrument also performed PCR and electrophoresis, when it did not (it prepared samples for those steps but did not perform them).

Reviewer #2: Aryan, Justin, Elio, Janet and M. Saad,

I enjoyed reading your recently submitted manuscript "OpenCell: A Low-cost, Open-Source, 3-in-1 device for DNA Extraction." The OpenCell device was clearly explained. The figures were well crafted and clear. I particularly appreciated the tests to select the optimal operating parameters for each function OpenCell provides, and the discussion of the cell lysis mechanism of the device. Bead-beating is one of the critical steps in many DNA extraction protocols, and the in-depth description of the bead-beating action in the article strengthens the credibility of OpenCell.

You performed your DNA extractions on spinach, with the Qiagen DNEasy Plant Pro kit. Qiagen kits are one of the most commonly used in DNA extraction, so it was nice to see this kit used. As you know, other types of DNA extractions from materials such as soil and water are common. I think that testing those sources of DNA with your device would be an interesting next step, and would enhance the paper itself, though this may be outside the scope of the current publication.

Before publication, I would like to see a gel of your extracted DNA before you perform your PCR, to compare the DNA size/smear with that of the commercial operation. I'd also like to see the PCR gel run with negative controls.

Minor grammar notes: in line 95, "All electrical connections can be made without via a solderless breadboard ... " delete the word "without." And should the Optimization and Characterization section be in past tense?

Overall I am happy to see this new device. I think it will open up DNA extraction to more scientists, especially - as the authors indicated - more traditionally non-science areas such as rural high schools and maker spaces. This is a great addition to the growing toolkit of open source, low-cost, accessible molecular biology devices. I recommend this paper for publication with minor revisions.

6. PLOS authors have the option to publish the peer review history of their article (what does this mean?). If published, this will include your full peer review and any attached files.

Reviewer #1: No

Reviewer #2: **Yes: **Emmett Smith

---

## [Editor Report · Decision Letter 1]

1 Feb 2024

OpenCell: A Low-cost, Open-Source, 3-in-1 device for DNA Extraction

PONE-D-23-30724R1

Dear Dr. Bhamla,

We’re pleased to inform you that your manuscript has been judged scientifically suitable for publication and will be formally accepted for publication once it meets all outstanding technical requirements.

Kind regards,

Sven Winter

Academic Editor

PLOS ONE